# Sodium–Glucose Cotransporter-2 Inhibitors Could Help Delay Renal Impairment in Patients with Type 2 Diabetes: A Real-World Clinical Setting

**DOI:** 10.3390/jcm11185259

**Published:** 2022-09-06

**Authors:** Gyunam Park, Byungha Choi, Soyoung Kang, Bomin Kim, Min Jung Chang

**Affiliations:** 1Department of Pharmacy and Yonsei Institute of Pharmaceutical Sciences, Yonsei University, Incheon 21983, Korea; 2Graduate Program of Industrial Pharmaceutical Science, Yonsei University, Incheon 21983, Korea; 3Department of Pharmaceutical Medicine and Regulatory Science, Yonsei University, Incheon 21983, Korea

**Keywords:** antidiabetic drug, renoprotective effect, sodium–glucose cotransporter-2 (SGLT2) inhibitor, diabetic kidney disease (DKD)

## Abstract

This study compared the renoprotective effects of sodium–glucose cotransporter-2 (SGLT2) inhibitors and dipeptidyl peptidase-4 (DPP-4) inhibitors in patients with type 2 diabetes mellitus (T2DM). We performed a retrospective cohort study using electronic medical records of patients with T2DM. The primary outcome was the first occurrence of an estimated glomerular filtration rate (eGFR) <45 mL/min/1.73 m^2^ after the index date. We analyzed changes in repeatedly measured laboratory data, such as eGFR and serum uric acid (SUA). We included 2396 patients (1198 patients in each group) in the present study. The rate of renal events was significantly lower in the SGLT2 inhibitors group than that in the DPP-4 inhibitors group (hazard ratio, 0.46; 95% CI, 0.29 to 0.72; *p* = 0.0007). The annual mean change in the eGFR was significantly smaller in the SGLT2 inhibitors group than that in the DPP-4 inhibitors group, with a between-group difference of 0.86 ± 0.18 mL/min/1.73 m^2^ per year (95% CI, 0.49 to 1.23; *p* < 0.0001). Moreover, the mean change in SUA was lower in the SGLT2 inhibitors group. Considering the lower incidence of renal impairment, the slower decline in eGFR, and reduced SUA, SGLT2 inhibitors could help delay renal impairment in patients with T2DM.

## 1. Introduction

The global burden of type 2 diabetes mellitus (T2DM) is steadily rising, presenting a major public healthcare challenge [1]. According to the International Diabetes Federation Diabetes Atlas, the global diabetes prevalence was estimated to be 9.3% (463 million people) in 2019 and is predicted to increase to more than 10% by 2030 [2]. Microvascular damage during diabetes is known to easily impact the kidneys [3]. Diabetic nephropathy (DN), more recently known as diabetic kidney disease (DKD), is the most frequent single cause of end-stage kidney disease (ESKD) [4].

Studies have demonstrated the benefits of angiotensin II receptor blockers (ARBs) and angiotensin-converting enzyme inhibitors (ACEIs) in diabetes-induced kidney damage [5,6]. Drugs that can mitigate the risk of DKD are urgently needed, and sodium–glucose cotransporter-2 (SGLT2) inhibitors have recently emerged. SGLT2 inhibitors differ from other antidiabetic agents owing to their insulin-independent glucose-lowering mechanisms [7].

SGLT2 inhibition can reportedly induce glucosuria and natriuresis. Notably, the expression and activity of renal glucose transporters, such as SGLT2, are higher in patients with T2DM than in non-T2DM individuals [8]. SGLT2 inhibitors also exhibit anti-inflammatory and antifibrotic effects on the kidneys [9]. Furthermore, SGLT2 inhibitors can decrease oxidative stress and inflammation, thereby contributing to a reduction in glomerulosclerosis [10]. SGLT2 inhibition can also attenuate glomerular hyperfiltration, which reportedly mediates DN and reduces albuminuria [11].

According to a recent review, the renoprotective effect of SGLT2 inhibitors has been demonstrated by several cardiovascular outcome trials and observational real-world evidence studies [12]. However, further studies are warranted to determine the effects of SGLT2 inhibitors on other renal outcomes in a real-world setting. Although some registry studies have been reported, data generated using electronic medical records (EMRs), including laboratory data, are lacking [13,14]. Moreover, the renoprotective effect of SGLT2 inhibitors in routine clinical practice remains poorly established in Asian cohorts. Accordingly, we evaluated the renoprotective effects of SGLT2 inhibitors in a real-world Asian cohort. We examined the prevention of early renal impairment in patients with T2DM.

Here, the goal was to establish the renoprotective effect of SGT2 inhibitors, for which an assessment/comparison was performed between SGLT2 inhibitors and dipeptidyl peptidase-4 (DPP-4) inhibitors. Furthermore, the effects of SGLT2 inhibitors and DPP-4 inhibitors on serum uric acid (SUA) are known to increase during chronic kidney disease (CKD) [15].

## 2. Materials and Methods

### 2.1. Data Source

The present retrospective cohort study was performed using EMRs of patients with T2DM who were treated between January 2014 and December 2019 at a single tertiary-level hospital in Seoul, Korea. The following data were collected from patient EMRs: demographic information such as age and sex; history of diagnoses and medical treatments; and laboratory data such as an estimated glomerular filtration rate (eGFR; from serum creatinine [SCr] levels), SUA, and glycated hemoglobin (HbA1c). In addition, medical records documenting renal replacement therapy, such as hemodialysis or peritoneal dialysis, were used to screen for severe renal impairment.

Baseline medical history data were collected one year prior to the index date. Diagnoses were classified using the International Classification of Disease, 10th Revision, Clinical Modification (ICD-10-CM) codes. Baseline laboratory data and medical treatments were collected six months prior to the index date. Medical treatments were classified according to the Anatomical Therapeutic Chemical code (Appendix A).

### 2.2. Study Population

We identified patients with T2DM aged 18 years or older who received at least 180 days or more of treatment with DPP-4 or SGLT2 inhibitors during the study period. The index date was defined as the date on which the patient first received either treatment. Patients were followed up until the first occurrence of any of the following: discontinuation of the index treatment, death, or study cut-off date (31 December 2019).

We assessed the renoprotective effect of SGLT2 inhibitors in patients with type 2 DM using DPP-4 inhibitors as an active comparator. Because placebo cannot be used in real-world practice, it was desirable to use a comparator, such as DPP-4 inhibitors, which have relatively neutral renal effects [16]. DPP-4 inhibitors were the most commonly prescribed secondary pharmacotherapy after metformin [17].

Patients who were co-administered DPP-4 and SGLT2 inhibitors during the study period were excluded. To include patients who visited the hospital regularly and underwent an appropriate follow-up, those with ˂80% drug compliance were excluded. Drug compliance was calculated as the number of prescription days divided by the period between the first and last prescription date.

Given that patients with an eGFR ˂45 mL/min/1.73 m^2^ cannot use SGLT2 inhibitors based on the product information provided by the Ministry of Food and Drug Safety (MFDS) in Korea, we excluded patients with renal impairment. Renal impairment was defined as follows: severe renal impairment (identified by end-stage renal disease (ESRD), hemodialysis or peritoneal dialysis, and kidney transplantation), CKD stage ≥3b (identified by an eGFR <45 mL/min/1.73 m^2^), or presence of macroalbuminuria (identified by a urine–albumin–creatinine ratio (UACR) ≥ 300 mg/g) based on the Kidney Disease: Improving Global Outcomes (KDIGO) guideline which classifies CKD based on cause, GFR, and albuminuria [18]. Patients with no baseline laboratory data (HbA1c or SCr levels) were excluded.

After applying the inclusion and exclusion criteria, the groups were matched at a 1:1 ratio, and 1198 patients were included in the SGLT2 inhibitors group and DPP-4 inhibitors group, respectively. The matching covariates were as follows: sex, age, eGFR, HbA1c, cardiovascular diseases (hypertension, dyslipidemia, myocardial infarction, stroke, heart failure, atrial fibrillation, peripheral artery disease), presence of diabetic complications (diabetic retinopathy, diabetic neuropathy, DN), cancer, and use of antidiabetic drugs (metformin, sulfonylurea, thiazolidinedione, other antidiabetic drugs (meglitinide or α-glucosidase inhibitors), insulin, glucagon-like peptide-1 receptor agonists), statins, other lipid-lowering drugs (ezetimibe or fibrates), calcium channel blockers, ACEIs, ARBs, β-blockers, thiazides, aldosterone antagonists, loop diuretics, antiplatelet agents (platelet aggregation inhibitors or P2Y12 inhibitors), and anticoagulants (warfarin or new oral anticoagulants).

### 2.3. Outcomes

Although some studies used an intention-to-treat approach, our analysis was conducted using an on-treatment approach to assess the effectiveness of the index treatment more accurately [14,18]. Follow-up was censored if patients discontinued the index treatment (SGLT2 inhibitors or DPP-4 inhibitors). Only outcomes that occurred while the patient received the index treatment were included in the analysis.

The primary outcome was the first occurrence of an eGFR ˂45 mL/min/1.73 m^2^ after the index date, which was assessed using time-to-event analysis. Several studies have used an eGFR ˂15 mL/min/1.73 m^2^ as the criterion to determine a renal event, which was used as the primary outcome [16,19,20]. However, we used the eGFR criterion of 45 mL/min/1.73 m^2^ to define renal events. The MFDS approval in Korea states that the use of SGLT2 inhibitors is not recommended for glycemic control when the eGFR is ˂45 mL/min/1.73 m^2^ (Appendix A). Among patients using SGLT2 inhibitors, no patient presented an eGFR ˂15 mL/min/1.73 m^2^ following an on-treatment approach in the present study. Another cohort study has reported no severe renal events in patients taking empagliflozin in Korea [13]. We performed subgroup analyses by sex, age, and eGFR groups to assess the interaction between treatment and subgroup, as well as modification of effect by subgroup.

To assess the additional effects of SGLT2 inhibitors and DPP-4 inhibitors, gradual changes in eGFR, and SUA from baseline were measured as secondary outcomes.

### 2.4. Statistical Analyses

All statistical analyses were performed using SAS version 9.4 (SAS Institute Inc., Cary, NC, USA) and the R program version 4.0.3 (The R Foundation for Statistical Computing, Vienna, Austria).

Baseline characteristics between the SGLT2 inhibitors group and DPP-4 inhibitors group were compared using chi-square tests for categorical variables and independent *t*-tests for continuous variables. Statistical significance was set at *p* < 0.05. We used the propensity score matching method to minimize the differences in baseline characteristics between the two groups. Patients were matched using the nearest-neighbor algorithm with a 1:1 ratio and caliper of 0.01. We considered covariates to be well-balanced if standardized mean differences between groups were ˂10%. Additionally, we confirmed the significance of *p*-values for all baseline covariates after matching.

Hazard ratios (HRs), 95% confidence intervals (95% CI), and *p*-values were calculated using Cox proportional hazard regression models for the primary outcome and subgroup analyses. HRs with 95% CI that did not include 1 were considered statistically significant. We conducted a survival analysis for each treatment group to estimate the risk of eGFR reduction. The time from the index date to renal events (eGFR <45 mL/min/1.73 m^2^) was assessed using the Kaplan–Meier (K–M) curve and log-rank test. A *p*-value of ˂0.05 was considered statistically significant for the log-rank test.

Additionally, we used the linear mixed model to analyze gradual changes in repeatedly measured eGFR values (calculated by the CKD-EPI equation) for the SGLT2 inhibitors group and DPP-4 inhibitors group. The change in eGFR from the baseline was designated as the dependent variable. Baseline values of eGFR (linear), visit time (linear), treatment group, and the interaction between visit and treatment groups were designated as fixed effects. The patient was designated to have a random effect. The visit periods (evaluation time points) were split at 1.5, 6, 12, 18, 24, 30, and 36 months (with an allowance of ±1.5). Patients with at least one value corresponding to a time point were included in the analyses. Thus, the number of patients at each time point differed and was less than the total number of matched pairs.

To evaluate the benefits of SGLT2 inhibitors, we measured the gradual changes in SUA in both treatment groups. Given the absence of any gradual difference in the slope between groups for SUA, we assumed that there was no interaction between time and group. Therefore, time (corresponding to visit) and the interaction term (corresponding to the interaction between visit and treatment group) were excluded from the fixed effect.

## 3. Results

### 3.1. Baseline Characteristics

Figure 1 presents a flowchart of patient inclusion in the study cohort. Before matching, 1210 and 10,441 users of SGLT2 inhibitors and DPP-4 inhibitors, respectively, met the study eligibility criteria. After matching, 2396 patients (1198 patients in each group) were included.

Statistical analyses were conducted before and after matching. After propensity score matching, the difference between the groups reduced, and no statistically significant differences were noted (Table 1). In addition, the standardized mean differences for all covariates were ˂10%, indicating that the groups were well-balanced (Table 1 and Appendix A).

### 3.2. Primary Outcome

The primary outcome was the first occurrence of renal events (defined as eGFR ˂45 mL/min/1.73 m^2^) after the index date. During the follow-up period, 101 renal events were documented in the study population. The number of renal events was significantly lower in the SGLT2 inhibitors group than that in the DPP-4 inhibitors group at 12.0 and 25.1 per 1000 person-years, respectively (HR, 0.46; 95% CI, 0.29 to 0.72; *p* = 0.0007) (Table 2). Figure 2 shows the cumulative incidence of renal events in the K–M curves. Notably, the incidence of renal impairment was significantly lower in the SGLT2 inhibitors group than that in the DPP-4 inhibitors group (log-rank *p* = 0.0005).

Subgroup analyses were conducted according to sex, age, and eGFR (Table 2). The renoprotective effect of SGLT2 inhibitors was generally consistent across examined subgroups. Following subgroup analyses by sex, age, and eGFR groups, no significant interactions were observed between the use of SGLT2 inhibitors and renal impairment (eGFR ˂ 45 mL/min/1.73 m^2^). On comparing sex-based differences, the HRs were 0.57 (95% CI, 0.32 to 1.03) in males and 0.34 (95% CI, 0.17 to 0.69) in females (*p* for interaction = 0.2755). HRs for patients aged < 65 years and elderly patients (≥65 years old) were 0.33 (95% CI, 0.16 to 0.69) and 0.66 (95% CI, 0.37 to 1.18), respectively (*p* for interaction = 0.1424). The eGFR was divided into the renal impairment group (eGFR < 90 mL/min/1.73 m^2^) and normal renal function group (eGFR ≥ 90 mL/min/1.73 m^2^). The HRs for patients in the renal impairment and normal groups were 0.51 (95% CI, 0.31 to 0.85) and 0.37 (95% CI, 0.12 to 1.11), respectively (*p* for interaction = 0.5804).

### 3.3. Secondary Outcomes

The least-squares mean (±standard error) change in the rate of eGFR decline was −1.13 ± 0.11 and −1.99 ± 0.15 mL/min/1.73 m^2^ per year in the SGLT2 inhibitors group and DPP-4 inhibitors group, respectively (Figure 3). This finding indicates that the annual mean change in eGFR in the SGLT2 inhibitors group was significantly smaller than that observed in the DPP-4 inhibitors group, with a between-group difference of 0.86 ± 0.18 mL/min/1.73 m^2^ per year (95% CI, 0.49 to 1.23; *p* < 0.0001).

Additionally, a reduction in SUA was seen in the SGLT2 inhibitors group (Appendix A). The least-squares mean (±SE) change in SUA in the SGLT2 inhibitors group was −0.34 ± 0.03 mg/dL, and it remained at a low level without continuous decline. On the other hand, the SUA of the DPP-4 inhibitors group was considerably elevated at 0.26 ± 0.03 mg/dL. The mean change in the SUA of the SGLT2 inhibitors group was lower than that of the DPP-4 inhibitors group, with a between-group difference of −0.59 ± 0.04 mg/dL (95% CI, −0.67 to −0.52; *p* < 0.0001).

## 4. Discussion

In the present study, we compared the renoprotective effects of SGLT2 inhibitors and DPP-4 inhibitors in patients with T2DM.

Based on evidence from large-scale randomized controlled trials (RCTs), dapagliflozin and canagliflozin were approved by the US Food and Drug Administration to reduce the risk of ESKD [21,22]. Numerous RCTs examining the renoprotective effect of SGLT2 inhibitors have primarily enrolled Caucasian patients; however, the present retrospective real-world study was conducted in an East Asian cohort. In addition, although some studies evaluated renal endpoints based solely on diagnosis, this study used laboratory data from EMRs. In addition, we evaluated the occurrence of an eGFR ˂45 mL/min/1.73 m^2^ as the primary endpoint, which is the main cut-off determining the use of SGLT2 inhibitors. 

Herein, the incidence of renal impairment was significantly lower in the SGLT2 inhibitors group than that in the DPP-4 inhibitors group (HR, 0.46; 95% CI, 0.29 to 0.72; *p* = 0.0007). The incidence of renal impairment in the SGLT2 inhibitors group was less than half during the last three-year observation period. In the subgroup analysis, females, patients aged <65 years, and patients with eGFR ˂90 mL/min/1.73 m^2^ in the SGLT2 inhibitors group exhibited reduced renal impairment when compared with those in the DPP-4 inhibitors group. 

In addition, the decline in eGFR was slower in the SGLT2 inhibitors group than that in the DPP-4 inhibitors group. Changes in eGFR values in the present study were within ranges observed in previous studies [18,23]. The SGLT2 inhibitors group showed a slower gradual decline in eGFR when compared with the DPP-4 inhibitors group, with a between-group difference of 0.86 ± 0.18 mL/min/1.73 m^2^ per year (95% CI, 0.49 to 1.23; *p* < 0.0001). We found an initial acute reduction in eGFR in the SGLT2 inhibitors group, which rapidly resolved. Previous studies have reported similar findings in those who were treated with SGLT2 inhibitors during the acute phase [18,19,20]. The acute phase is typically observed when patients first initiate SGLT2 inhibitors, and eGFR subsequently stabilized [24]. The acute phase is probably caused by the increased delivery of sodium and glucose to the macula densa due to SGLT2 blockade, thereby resulting in renal tubuloglomerular feedback and subsequent afferent vasoconstriction [25]. 

The benefits of SGLT2 inhibitors can be attributed to glycosuria and natriuresis, which result in increased sodium intake and activation of renal tubuloglomerular feedback [26]. This mechanism can reduce intraglomerular hypertension, a common pathological condition in DKD. SGLT2 inhibitors have proven to be effective in preventing CKD progression, and recent American Diabetes Association guidelines were updated to recommend SGLT2 inhibitors to delay CKD progression in patients with CKD ≥3 [27]. In the present study, the results indicate that the use of SGLT2 inhibitors reduced the incidence of early renal impairment (especially eGFR <45 mL/min/1.73 m^2^). Considering the reduced incidence of early renal impairment and delayed eGFR decline, renoprotective effects can be expected when treated during the early stages before CKD development. Therefore, SGLT2 inhibitors could be prescribed during the early stages of CKD to reduce progression. 

SGLT2 inhibitors also lowered serum uric acid (SUA) more than DPP-4 inhibitors did in this study. The effect of lowering SUA is relatively well evidenced [28,29,30], and the mechanism is not clear, but it is considered to be related to several uric acid transporters [31,32]. This reduction in SUA can be beneficial in terms of the cardio-renal effect [33]. Elevated SUA is associated with an increased risk of cardiovascular disease [34,35]. Moreover, it can be a risk factor for the development and progression of CKD [15,36]. Therefore, lowering SUA by SGLT2 inhibitors can reduce cardiovascular (CV) events and slow the progression of CKD.

The limitations of the present study need to be addressed. The definition of renal events used as the primary outcome in the current study differed from that used in previous studies [16,20,37]. In Korea, SGLT2 inhibitors are not recommended for glycemic control in patients with T2DM presenting an eGFR ˂45 mL/min/1.73 m^2^, and we included these patients per se. However, other studies used the intention-to-treat approach to define renal events as severe renal dysfunction such as ESRD or eGFR ˂15 mL/min/1.73 m^2^. Therefore, it was challenging to identify new drug users in the treatment groups owing to the study design. Studies using health insurance or registered cohorts encountered less difficulty identifying new users because all prescription details were provided by health institutions [13,16,18]. However, the current study was a retrospective study at a single tertiary-level hospital; therefore, prescriptions provided by other hospitals could not be identified. Despite these limitations, this study used EMRs, including laboratory data, to evaluate the renoprotective effect of SGLT2 inhibitors in Korea with real-world clinical experience.

## 5. Conclusions

This retrospective study compared the renoprotective effect of SGLT2 inhibitors and DPP-4 inhibitors, and SGLT2 inhibitors were associated with a lower incidence of renal impairment. Moreover, SGLT2 inhibitors showed a slower decline in eGFR and reduced SUA. Therefore, SGLT2 inhibitors could help delay renal impairment in patients with T2DM. This study confirms the renoprotective effect of SGLT2 inhibitors and provides real-world evidence of their renal benefits in East Asian patients with T2DM.

## Figures and Tables

**Figure 1 jcm-11-05259-f001:**
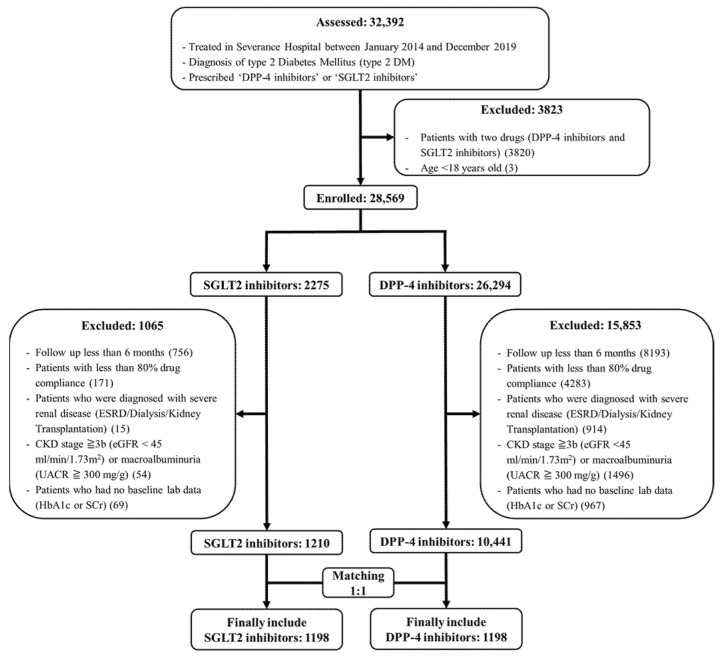
Flowchart of patient inclusion and exclusion. CKD, chronic kidney disease; DPP-4 dipeptidyl peptidase-4; eGFR, estimated glomerular filtration rate; ESRD, end-stage renal disease; HbA1c, glycated hemoglobin; SCr, serum creatinine; SGLT2, sodium–glucose cotransporter-2; UACR, urine–albumin–creatinine ratio.

**Figure 2 jcm-11-05259-f002:**
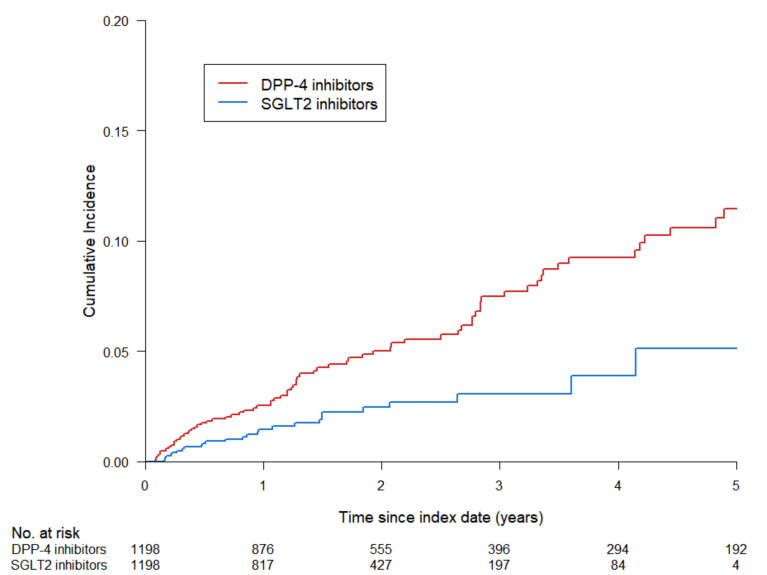
Comparison of cumulative incidence of renal events (eGFR < 45 mL/min/1.73 m^2^) in matched cohort between SGLT2 inhibitors group and DPP-4 inhibitors group. eGFR, estimated glomerular filtration rate; SGLT2, sodium–glucose cotransporter-2; DPP-4, dipeptidyl peptidase-4.

**Figure 3 jcm-11-05259-f003:**
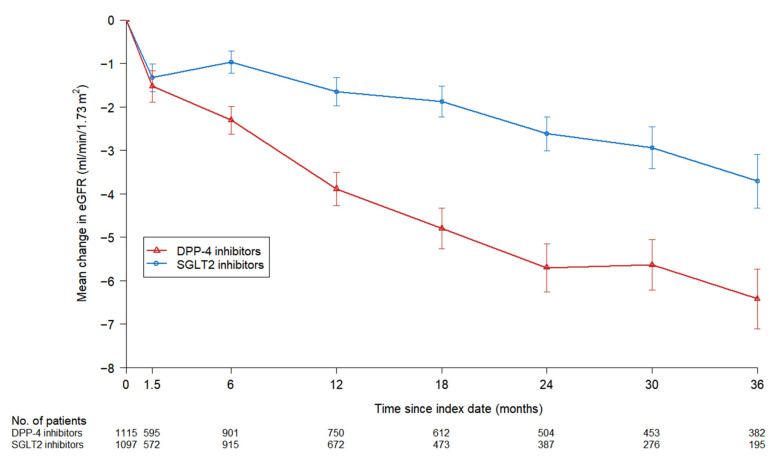
Change in eGFR in the SGLT2 inhibitors group and DPP-4 inhibitors group. Error bars indicate standard error. The numbers below the graph refer to the number of patients at each time point. eGFR, estimated glomerular filtration rate; SGLT2, sodium–glucose cotransporter-2; DPP-4, dipeptidyl peptidase-4.

**Table 1 jcm-11-05259-t001:** Baseline characteristics of propensity-score-matched pairs of the SGLT2 inhibitors group and DPP-4 inhibitors group.

	SGLT2 Inhibitors (*n* = 1198)	DPP-4 Inhibitors (*n* = 1198)	*p*-Value	Standardized Mean Difference *
Age (years), mean (SD)	55.4 (12.1)	54.9 (13.2)	0.3971	3.7%
Age group (years), *n* (%)			0.3825	-
<65	931 (77.7)	913 (76.2)		
≥65	267 (22.3)	285 (23.8)		
Sex, *n* (%)			0.9665	0.2%
Male	737 (61.5)	736 (61.4)		
Female	461 (38.5)	462 (38.6)		
HbA1c (%), mean (SD)	7.86 (1.52)	7.88 (1.72)	0.7845	1.2%
eGFR (mL/min/1.73 m^2^), mean (SD)	96.1 (16.3)	96.6 (17.7)	0.4285	3.3%
eGFR group (mL/min/1.73 m^2^), *n* (%)			0.4764	-
<60	35 (2.9)	45 (3.8)		
60–90	318 (26.5)	306 (25.5)		
≥90	845 (70.5)	847 (70.7)		
Serum uric acid ^†^ (mg/dL), mean (SD)	4.99 (1.29)	4.91 (1.33)	0.1602	-
Medical history, *n* (%)				
Hypertension	708 (59.1)	705 (58.9)	0.9008	0.5%
Dyslipidemia	515 (42.6)	500 (41.7)	0.7720	1.2%
Myocardial infarction	59 (4.9)	59 (4.9)	1.0000	0.0%
Stroke	43 (3.6)	49 (4.1)	0.5235	2.3%
Heart failure	67 (5.6)	65 (5.4)	0.8579	0.8%
Atrial fibrillation	41 (3.4)	43 (3.6)	0.8242	0.8%
Peripheral artery disease	64 (5.3)	65 (5.4)	0.9279	0.4%
Diabetic retinopathy	248 (20.7)	236 (19.7)	0.5415	2.6%
Diabetic neuropathy	370 (30.9)	339 (28.3)	0.1653	6.1%
Diabetic nephropathy	83 (6.9)	81 (6.8)	0.8715	0.7%
Cancer	122 (10.2)	127 (10.6)	0.7378	1.2%
Treatments, *n* (%)				
Metformin	1116 (93.2)	1120 (93.5)	0.7434	1.1%
Sulfonylurea	313 (26.1)	321 (26.8)	0.7110	1.5%
Thiazolidinedione	139 (11.6)	146 (12.2)	0.6587	1.8%
Other antidiabetic drugs	21 (1.8)	21 (1.8)	1.0000	0.0%
Insulin	289 (24.1)	291 (24.3)	0.9240	0.4%
GLP-1 receptor agonists	6 (0.5)	5 (0.4)	0.7625	1.4%
Statins	755 (63.0)	765 (63.9)	0.6714	1.7%
Other lipid-lowering drugs	223 (18.6)	225 (18.8)	0.9165	0.5%
Calcium channel blockers	357 (29.8)	351 (29.3)	0.7882	1.1%
ACEIs	114 (9.5)	127 (10.6)	0.3772	3.9%
ARBs	469 (39.2)	462 (38.6)	0.7692	1.2%
β-blockers	300 (25.0)	296 (24.7)	0.8501	0.8%
Thiazides	96 (8.0)	97 (8.1)	0.9402	0.3%
Aldosterone antagonist	71 (5.9)	77 (6.4)	0.6106	2.2%
Loop diuretics	102 (8.5)	106 (8.9)	0.7716	1.1%
Antiplatelet agents	485 (40.5)	495 (41.3)	0.6778	1.7%
Anticoagulants	43 (3.6)	41 (3.4)	0.8242	0.9%

Percentages may not total 100 because of rounding. ^†^ Serum uric acid was evaluated for 1189 users of SGLT2 inhibitors and 1190 users of DPP-4 inhibitors and was not used as a matching variable. * Standardized mean difference (%) of ˂10% was considered negligible. SGLT2, sodium–glucose cotransporter-2; DPP-4, dipeptidyl peptidase-4; HbA1c, glycated hemoglobin; eGFR, estimated glomerular filtration rate; Other antidiabetic drugs, meglitinide or α-glucosidase inhibitors; GLP, glucagon-like peptide; Other lipid-lowering drugs, ezetimibe or fibrates; ACEI, angiotensin-converting enzyme inhibitor; ARB, angiotensin II receptor blocker; Antiplatelet agents, platelet aggregation inhibitors or P2Y12 inhibitors; Anticoagulants, warfarin or new oral anticoagulants.

**Table 2 jcm-11-05259-t002:** Subgroup analyses of renal events for SGLT2 inhibitors group compared with DPP-4 inhibitors group.

Subgroups	SGLT2 Inhibitors	DPP-4 Inhibitors	Hazard Ratio (95% CI)		*p* Value for Interaction
	No. of Patients/Total No.				
Total	26/1198	75/1198	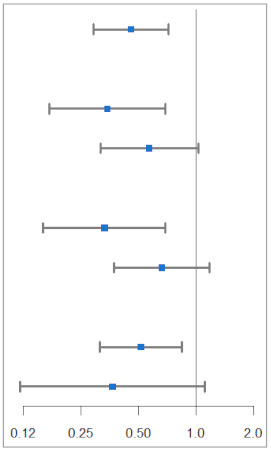	0.46 (0.29–0.72)	
Sex					0.2755
Female	10/461	37/462		0.34 (0.17–0.69)	
Male	16/737	38/736		0.57 (0.32–1.03)	
Age (years)					0.1424
<65	9/931	34/913		0.33 (0.16–0.69)	
≥65	17/267	41/285		0.66 (0.37–1.18)	
eGFR (mL/min/1.73 m^2^)					0.5804
<90	22/353	61/351		0.51 (0.31–0.85)	
≥90	4/845	14/847		0.37 (0.12–1.11)	


SGLT2: sodium–glucose cotransporter-2; DPP-4: dipeptidyl peptidase-4.

## Data Availability

All the re-identified data are available upon reasonable request (mjchang@yonsei.ac.kr).

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
