# Peer review of "Sodium–Glucose Cotransporter-2 Inhibitors Could Help Delay Renal Impairment in Patients with Type 2 Diabetes: A Real-World Clinical Setting"

_jcm, 2022, doi:10.3390/jcm11185259_

Round 1

Reviewer 1 Report

SGLT2i is an effective tool in dealing with DKD, but it has been overly used in the clinical setting now. This study proposed to compare the efficacy of SGLT2i and DPP-4 in preserving kidney function in the 2-DM patient. Overall, the study was well designed, and some data was interesting. The major concern is the rationale of sample size determination. Since it is a retro-cohort, all cases that fit the criteria should be included for further analysis. The authors do not need to do a 1:1 match. Otherwise, it will cause potential misleading. This point should be satisfactorily addressed. 

Author Response

Dear the reviewer,

We appreciate for the kind and specific comments of yours.

We carefully answered the comments and revised the manuscript, and the revised manuscript was highlighted (Please see the attachment).

Min Jung Chang

Reviewer 1

Point 1: SGLT2i is an effective tool in dealing with DKD, but it has been overly used in the clinical setting now. This study proposed to compare the efficacy of SGLT2i and DPP-4 in preserving kidney function in the 2-DM patient. Overall, the study was well designed, and some data was interesting. The major concern is the rationale of sample size determination. Since it is a retro-cohort, all cases that fit the criteria should be included for further analysis. The authors do not need to do a 1:1 match. Otherwise, it will cause potential misleading. This point should be satisfactorily addressed.

Response 1: In the case of retrospective studies, baseline differences between the two groups may occur much larger than those of randomized controlled studies. So, we used the propensity score matching method to minimize differences in baseline characteristics between SGLT2 inhibitors and DPP-4 inhibitors groups. Through this matching, the two groups could be matched almost similarly for several factors that affected the outcomes. All matching covariates such as sex, age, eGFR, HbA1c, medical history and treatments are properly balanced, and bias were significantly reduced by matching variables that are thought to directly associated with renal events, especially age, eGFR, and diabetic complications.

Also, to assess the renoprotective effects of SGLT2 inhibitors and DPP-4 inhibitors, changes in repeatedly measured eGFR over time were measured as secondary outcomes. It was expected that the bias could be greatly reduced by comparing the change of eGFR after matching a similar eGFR baseline between the two groups through matching.

When using propensity score matching method, a sufficient number of subjects is required as a comparative group, and DPP-4 inhibitors were the most suitable for comparator. Also, 1:1 is the most common ratio in the propensity score matching method.

For the better understanding, we add the reason to select DPP-4 inhibitors as the comparator. (Line 90-94: We assessed the renoprotective effect of SGLT2 inhibitors in patients with type 2 DM using DPP-4 inhibitors as an active comparator. Because placebo cannot be used in a real-world practice, it was desirable to use a comparator, such as DPP-4 inhibitors, which have relatively neutral renal effect [16]. DPP-4 inhibitors were the most commonly prescribed secondary pharmacotherapy after metformin [17].)

Reviewer 2 Report

Title: The reno-protective effect of sodium-glucose cotransporter-2 inhibitors in patients with type 2 diabetes: a real-word retrospective cohort study 

Comments: The study is very intricate and delicately presented. It comprehensively explores the two treatments: DPP-4 and SGLT2 inhibitors. The study is very extensively conducted presenting with promising results. However, a few grammatical errors were noticed for reviewing and the title could be worded to be a bit more conclusive to the inhibitors under investigation. 

1.    The clinical relevance and focus of study are described

2.    Comprehensive literature review is conducted including the exploration of both the treatment with DPP-4 and SGLT2 inhibitors.

3.    Methodology is extensive, screening and selection process were reported with detail.

a.    All the stages and definitions and values used for classification of outcome were cited  

4.    Evaluation of results was detail. The figures and tables are self-explanatory and presented in an easy to understand and systematic manner

5.    Please review the manuscript, there are some grammatical errors – line 87.

6.    Reword the title of the manuscript

7.    Conclusion; adequate summary of the study

Author Response

Dear the reviewer,

We appreciate for the kind and specific comments of yours.

We carefully answered the comments and revised the manuscript, and the revised manuscript was highlighted (Please see the attachment).

Min Jung Chang

Reviewer 2

The study is very intricate and delicately presented. It comprehensively explores the two treatments: DPP-4 and SGLT2 inhibitors. The study is very extensively conducted presenting with promising results. However, a few grammatical errors were noticed for reviewing and the title could be worded to be a bit more conclusive to the inhibitors under investigation.

Point 1: The clinical relevance and focus of study are described

Point 2: Comprehensive literature review is conducted including the exploration of both the treatment with DPP-4 and SGLT2 inhibitors.

Point 3: Methodology is extensive, screening and selection process were reported with detail.

  1. All the stages and definitions and values used for classification of outcome were cited

Point 4: Evaluation of results was detail. The figures and tables are self-explanatory and presented in an easy to understand and systematic manner

Response 1-4: We appreciate for your kind and structured comments. We will carefully revise the manuscript as you recommended.

Point 5: Please review the manuscript, there are some grammatical errors – line 87.

Response 5: We apologize for our errors. (Line 88: c) We corrected some grammatical errors including this one. Please see our updated manuscript.

Point 6: Reword the title of the manuscript

Response 6: We agree with the reviewer’s comment. We revised our title considering the reviewer’s comments to “The sodium-glucose cotransporter-2 inhibitors could help delay renal impairment in patients with type 2 diabetes: a real-world clinical setting”.

Point 7: Conclusion; adequate summary of the study

Response 7: We agree with the reviewer’s comment. We revised our abstract and conclusions considering the reviewer’s comments (Line 22-27, 325-330).

After 7

(Abstract, Line 22-27) The annual mean change in the eGFR was significantly smaller in the SGLT2 inhibitors group than that in the DPP-4 inhibitors group, with a between-group difference of 0.86 ± 0.18 mL/min/1.73 m2 per year (95% CI, 0.49 to 1.23; p<0.0001). Also, the mean change in SUA was lower in the SGLT2 inhibitors group. Considering the lower incidence of renal impairment, slower decline in eGFR, and reduced SUA, SGLT2 inhibitors could help delay renal impairment in patients with T2DM.

(Conclusions, Line 325-330): This retrospective study compared the renoprotective effect of SGLT2 inhibitors and DPP-4 inhibitors, and SGLT2 inhibitors associated with a lower incidence of renal impairment. Also, SGLT2 inhibitors showed a slower decline in eGFR and reduced SUA. Therefore, SGLT2 inhibitors could help delay renal impairment in patients with T2DM. This study confirms the renoprotective effect of SGLT2 inhibitors and provides real-world evidence of their renal benefits in East Asian patients with T2DM.

Round 2

Reviewer 1 Report

The authors addressed my concerns.